# A prebiotic template-directed peptide synthesis based on amyloids

Saroj K. Rout[1], Michael P. Friedmann[1], Roland Riek[1] & Jason Greenwald[1]

The prebiotic replication of information-coding molecules is a central problem concerning life's origins. Here, we report that amyloids composed of short peptides can direct the sequence-selective, regioselective and stereoselective condensation of amino acids. The addition of activated DL-arginine and DL-phenylalanine to the peptide RFRFR-NH$_2$ in the presence of the complementary template peptide Ac-FEFEFEFE-NH$_2$ yields the isotactic product FRFRFRFR-NH$_2$, 1 of 64 possible triple addition products, under conditions in which the absence of template yields only single and double additions of mixed stereochemistry. The templating mechanism appears to be general in that a different amyloid formed by (Orn) V(Orn)V(Orn)V(Orn)V-NH$_2$ and Ac-VDVDVDVDV-NH$_2$ is regioselective and stereo-selective for N-terminal, L-amino-acid addition while the ornithine-valine peptide alone yields predominantly sidechain condensation products with little stereoselectivity. Furthermore, the templating reaction is stable over a wide range of pH (5.6–8.6), salt concentration (0–4 M NaCl), and temperature (25–90 °C), making the amyloid an attractive model for a prebiotic peptide replicating system.

[1] Laboratory of Physical Chemistry, ETH Zürich, Vladimir-Prelog-Weg 2, 8093 Zürich, Switzerland. Correspondence and requests for materials should be addressed to R.R. (email: roland.riek@phys.chem.ethz.ch) or to J.G. (email: jason.greenwald@phys.chem.ethz.ch)

Once thought to be just an aberration of biology, amyloids with their unique cross-β fold are now accepted as an integral structural and functional entity in all forms of life[1,2]. Furthermore, amyloid fibers composed of short peptides can comprise a variety of enzyme-like catalytic activities[3–7]. Based on these findings, we and others have hypothesized that such amyloid catalysts may have played an important role in prebiotic molecular evolution[2,8–11]. Our recent finding that amyloid fibers arise spontaneously from amino acids under prebiotic conditions supported this hypothesis and opened up the possibility that peptides, acting as both an informational and catalytic entity, preceded life on Earth[12]. So, in addition to being orders of magnitude more stable than phosphodiester-based nucleic acid polymers, it appears that amyloids have a trivial abiotic synthesis route compared to the complexity of known catalytic RNAs. Despite the relative simplicity of the peptides that can form catalytic amyloids, the question remains as to how a specific peptide sequence could become significantly represented in a system of polymerizing amino acids or, more specifically, whether amyloids can replicate themselves using simple chemical building blocks. To address this question, we set out to determine to what extent an amyloid can act as a catalyst or template in the synthesis of its constituent peptides.

The conformational templating or seeding effect that amyloids exert in the conversion of soluble peptides and proteins into their insoluble β-aggregate form is well documented and underlies the molecular mechanism of many amyloid-related diseases. This conformational replication is analogous to crystallization and is inherent to the highly ordered structure of the aggregate[1,13]. In terms of chemical replication, previous studies have shown that a peptide amyloid can act as a template to accelerate the rate of condensation (via native chemical ligation) of two smaller peptides that comprise the amyloidogenic peptide[14,15]. Very recently, this concept was extended to include an interconverting network of 4 regio- and stereoisomers in which the parent amyloid acted as a template to enhance the yield of self-products at the expense of kinetically more favorable products[16]. Considering the structure of the β-sheet[17,18] it is clear that as a template, an amyloid should be stereoselective. An early experimental demonstration of the stereoselective nature of β-aggregates was given by Brack and Spach[19]. They found that the formation of a β-structured aggregate from aggregation-prone sequences requires peptides that have at least a minimal continuous stretch of homochiral (isotactic) amino-acid residues. Building upon this and the work of Luisi and coworkers[20] on the polycondensation of racemic amino acids, the discovery by Lahav and coworkers that the condensation of activated racemic amino acids produces a precipitate in which the peptides display a very high degree of isotacticity[21] remains very intriguing. The authors conjecture that short peptides aggregate into thermodynamically favored racemic rippled β-sheets early in the reaction, and that these subsequently act as a template for the further addition of activated monomers.

However, concerning questions of abiogenesis, there is still little experimental evidence of molecular replication in a truly prebiotic system. While the templating capability of the amyloid structure is well documented, it remains to be shown how far their self-organizing nature can be taken towards a self-replicating system of simpler molecules in more prebiotic conditions. With this in mind we designed peptides with binary alternating sequences of hydrophilic and hydrophobic amino-acid residues, a pattern known to favor amphipathic β–strands that aggregate into cross-β fibers[22–24]. The individual template and substrate peptides were designed to be soluble at neutral pH due to charge repulsion, yet able to form an amyloid when mixed. We have investigated four different template/substrate pairs and in all cases observed a significant enhancement of sequence-selective and stereoselective amino-acid addition. Furthermore, in a substrate peptide that involved both side-chain and N-terminal reactive sites, we also observed a regioselectivity for N-terminal additions in the substrate/template amyloid that was absent in the soluble substrate.

## Results

**Selective addition of amino acids to $R(FR)_3$ templated by $(FE)_4$.** Throughout this study, we use substrate peptides (analogous to primer strands in DNA replication) that are the charge complement of their template peptide. For example, the template $(FE)_4$ (Ac-FEFEFEFE-NH$_2$, Table 1) has a net negative charge at neutral pH, while substrate $R(FR)_3$ (RFRFRFR-NH$_2$) has a net positive charge and one phenylalanine residue less than the template (Fig. 1a). Additionally, the template peptides have an N-terminal acetylation and all peptides have a C-terminal amide. Both $(FE)_4$ and $R(FR)_3$ are soluble separately above 1 mM (Supplementary Fig. 1). However, a mixture of 100 µM of each peptide in 50 mM phosphate at pH 7.4 yields a precipitate that has many hallmarks of an amyloid including an intense amide C = O stretch at 1625 cm$^{-1}$ in its infrared spectrum and fibril morphology in transmission electron micrographs (Supplementary Figs. 1 and 2). Phenylalanine was activated with carbonyldiimidazole (CDI) and then added in stoichiometric amounts to the $R(FR)_3/(FE)_4$ aggregate or to soluble $R(FR)_3$. After 18 h, the reactions were solubilized with guanidine and analyzed by reverse-phase HPLC. Surprisingly, the yield of the phenylalanine addition product $(FR)_4$ was much higher in the aggregated sample (51%), compared to the soluble control (2.7%; Fig. 1b). The low yield in the soluble reaction was not simply due to a non-reactive soluble peptide: we also tested four other amino acids (D, G, V and R, Fig. 1c–f) similarly activated with CDI, and $R(FR)_3$ reacts with aspartate to form $DR(FR)_3$ (33% yield) nearly as efficiently as does $R(FR)_3/(FE)_4$ (35% yield, Fig. 1c). Interestingly, despite the efficiency of phenylalanine addition to the co-aggregate, double additions of phenylalanine were not detected in the stochiometric reaction. While glycine did yield double additions, its reaction with $R(FR)_3/(FE)_4$ strongly favored single additions relative to soluble $R(FR)_3$ (Fig. 1d). Taking additionally into account the outcomes of the valine and arginine reactions (Fig. 1 and Supplementary Table 1) these results indicate that the $R(FR)_3/(FE)_4$ co-aggregated amyloid is sequence-specific for the addition of hydrophobic amino acids, enhancing only single additions to the substrate.

Thus, the reaction on the amyloid is not simply enhanced by an increase in the nucleophilicity of the N-terminal amine, but is

---

### Table 1 Selected peptides used in this study

| Name | Composition[a] | Source[b] |
|---|---|---|
| $(FE)_4$ | Ac-FEFEFEFE-NH$_2$ | SPPS |
| $R(FR)_3$ | RFRFRFR-NH$_2$ | SPPS & reaction product |
| $(FR)_3$ | FRFRFR-NH$_2$ | SPPS & reaction product |
| $R(FR)_2$ | RFRFR-NH$_2$ | SPPS |
| $(FR)_4$ | FRFRFRFR-NH$_2$ | Reaction product |
| $FF(FR)_3$ | FFFRFRFR-NH$_2$ | Reaction product |
| $V(DV)_4$ | Ac-VDVDVDVDV-NH$_2$ | SPPS |
| $(OV)_4$ | OVOVOVOV-NH$_2$ | SPPS |
| $V(OV)_4$ | VOVOVOVOV-NH$_2$ | SPPS & reaction product |

[a] The standard single letter code for amino acids is used; O is for ornithine, Ac- is for acetylated N-terminus and –NH$_2$ for amidated C-terminus
[b] SPPS peptides were synthesized as described in the methods. The tabulated reaction products only include those most discussed in the manuscript

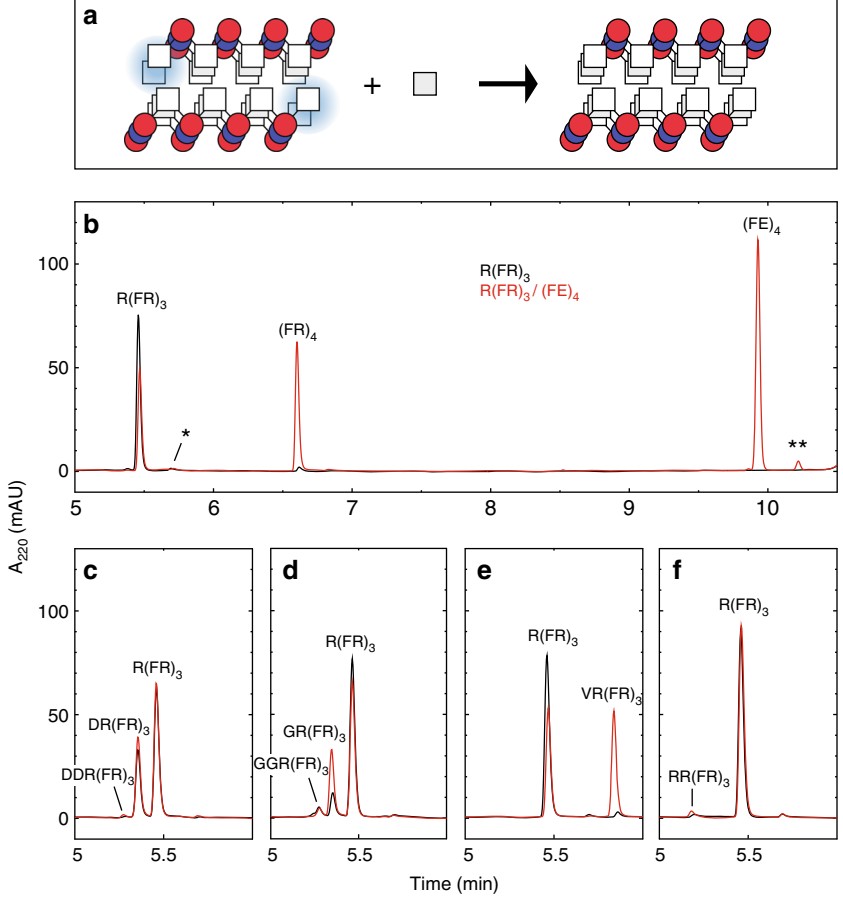

**Fig. 1** Single addition reactions with the $R(FR)_3/(FE)_4$ amyloid. **a** Schematic model of the amyloid-templated addition reaction, depicting just three layers of the repetitive amyloid structure in which the phenylalanine is represented by squares and the arginine and aspartate by blue and red circles respectively. The blue shading highlights the sites of addition in the substrate peptide. **b**–**f**, Reverse-phase HPLC chromatograms of the addition reactions with phenylalanine (**b**), aspartate (**c**), glycine (**d**), valine (**e**) and arginine (**f**). The * and ** indicate the C-terminal deamidated substrate and template, respectively, that remained after purification. The reactions with just the substrate $R(FR)_3$ are the black traces and those with the amyloid $R(FR)_3/(FE)_4$ are the red ones. The $R(FR)_3$ concentration was 100 μM, the $(FE)_4$ concentration was 130 μM and amino acids were all at 100 μM

influenced by the physicochemical properties of the incoming amino acid. Indeed, in the context of the $R(FR)_3/(FE)_4$ amyloid, the addition of DL-phenylalanine is highly stereoselective for the L-enantiomer (Fig. 2). In the range of phenylalanine concentrations that we tested (25–200 μM), there is a distinct dependence of the product diastereomeric excess (d.e.) on the phenylalanine concentration. This would suggest that the reaction kinetics are not simply second order with distinct rate constants for L and D additions, but rather that preceding the condensation there is a binding event, perhaps with the *N*-[imidazolyl-(1)-carbonyl]-phenylalanine or *N*-carboxyanhydride intermediate, whose affinity is in the micromolar range. The stereoselectivity of the $R(FR)_3/(FE)_4$ amyloid for five additional amino acids (A, V, L, Y and W) was similarly tested and found to be consistently L-selective, concentration dependent, and yielding higher product d.e. than the soluble $R(FR)_3$ peptide (Supplementary Fig. 3, Supplementary Table 2). As expected for the reaction between two chiral molecules, the addition of amino acids to soluble peptide displays some stereoselectivity albeit to a much lesser extent. In fact, soluble $R(FR)_3$ favors the D-enantiomers of leucine, phenylalanine, tyrosine and tryptophan, such that for these amino acids the $R(FR)_3/(FE)_4$ amyloid has inverted and enhanced stereoselectivity relative to soluble $R(FR)_3$. The largest increases in stereoselectivity for the amyloid relative

to the soluble substrate occurred for phenylalanine and valine: for example, at 200 μM amino acid (100 μM each enantiomer) the d.e. of phenylalanine addition products increased from 0% to 71% and for valine from −3 to 76%. As expected, the stereoselectivity for L-additions produces an excess of D-enantiomers in the unreacted pool of amino acid (Supplementary Fig. 4).

**Kinetics of amyloid-templated additions**. For an initial assessment of the reaction kinetics, we measured the time dependence of product formation at our standard conditions (100 μM activated phenylalanine) over a period of 24 h. The results depicted in Supplementary Fig. 5 show that the majority of the $(FR)_4$ product is formed within 30 min, with the yield peaking around 3 h and then sinking about 20%, primarily due to multiple additions. In an effort to get a more detailed understanding of the reaction mechanism we measured the early kinetics of the reaction for both $R(FR)_3/(FE)_4$ and the soluble $R(FR)_3$ at various L- and D-phenylalanine concentrations (Supplementary Figs. 6–8). The additions of L- or D-phenylalanine to soluble peptide appear to be first order with respect to phenylalanine (Supplementary Fig. 8c, d) as would be expected for a simple uncatalyzed addition. In contrast, the reaction kinetics of the

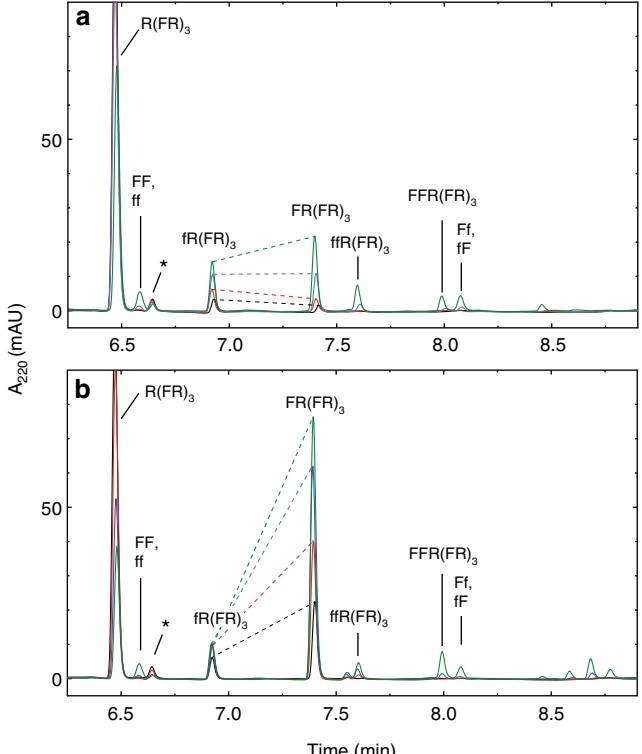

**Fig. 2** Stereospecificity of the single additions to $R(FR)_3/(FE)_4$ amyloid. Reverse-phase HPLC chromatograms of the addition reactions of DL-phenylalanine with soluble substrate $R(FR)_3$ (**a**) and amyloid $R(FR)_3/(FE)_4$ (**b**). The $R(FR)_3$ concentration was 100 μM, the $(FE)_4$ concentration was 130 μM and amino acids were at 25 μM (black), 50 μM (red), 100 μM (blue), 200 μM (green) of each enantiomer. The D-amino acid addition products are denoted with lower case letters. The single addiction product diastereomers are connected with a dashed line to illustrate differences in the yield of the products at different concentration of amino acid. The * indicates the C-terminal deamidated substrate that remained after purification. For a qualitative comparison of the specificity of addition of other amino acids, see Supplementary Fig. 3 and Supplementary Table 2

L-additions to the $(FR)_4/(FE)_4$ amyloid is much faster and of greater complexity, including a burst phase (Supplementary Figs. 7 and 8a). The data fit neither a first order reaction nor a Michaelis Menten-type mechanism (assuming a turn-over of 1). This complexity is specific for L-additions, since the additions of D-phenylalanine to $(FR)_4/(FE)_4$ appear to be first order with respect to phenylalanine (Supplementary Fig. 8b) (for some further details we refer the reader to the Supplementary Discussion).

**Consecutive additions to amyloid-templated substrates.** To test the sequence specificity and consecutive templating capability of the amyloids, we designed the shorter substrate $(FR)_3$ (FRFRFR-NH$_2$), now two residues shorter than the template $(FE)_4$. After demonstrating that the $(FR)_3/(FE)_4$ mixture also formed amyloid aggregates (Supplementary Figs. 1 and 2) we performed stoichiometric addition reactions with phenylalanine, glycine, arginine, valine, or aspartate. Instead of selecting the hydrophobic amino acids phenylalanine and valine, this amyloid specifically enhances condensation with arginine. Four times more arginine addition occurred with the $(FR)_3/(FE)_4$ co-aggregate than with the $(FR)_3$ soluble peptide, while the other amino acids reacted at least twice as much with the soluble peptide than with the co-aggregate (Fig. 3a). Even though the amyloid diminished their addition to the substrate, phenylalanine and valine actually yielded double addition products in reactions with the amyloid, consistent with the previous result (Fig. 1, Supplementary Table 1) that the template favors a hydrophobic amino acid for this position of the substrate. In fact, in a mixture of phenylalanine and arginine (100 μM each), $(FR)_3/(FE)_4$ reacted to form the sequence-specific double addition product $(FR)_4$ (3.4%) as well as the double phenylalanine addition product FF $(FR)_3$ (3.9%) while the soluble $(FR)_3$ yielded only single additions of one or the other amino acid: arginine (1.9%), or phenylalanine (14%) (Fig. 3b).

With a yet shorter substrate, $R(FR)_2$ we wanted to test the amyloid templating for three consecutive residues, however, the $R(FR)_2/(FE)_4$ mixture remained soluble with no obvious amyloid signatures (Supplementary Figs. 1 and 2). However, the

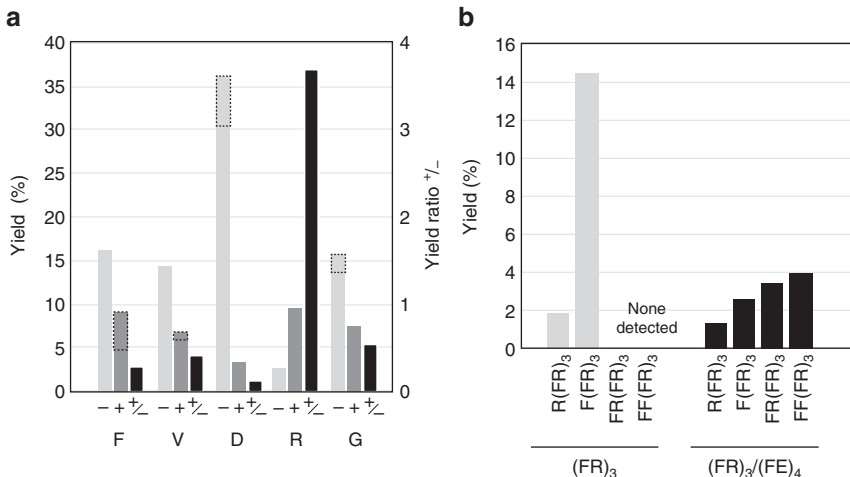

**Fig. 3** Sequence-specific, consecutive double addition reactions with the $(FR)_3/(FE)_4$ amyloid. **a** Product yields in addition reactions with the individual amino acids phenylalanine, valine, aspartate, arginine and glycine. The yields for the reactions without template are in light gray, with template in gray and their ratio plotted on a different scale (right axis) is in black. The double addition products are outlined with the dotted lines. **b** Product yields in a mixture of arginine and phenylalanine. For all reactions, the $(FR)_3$ concentration was 100 μM, the $(FE)_4$ concentration was 130 μM and individual amino acids were all at 100 μM

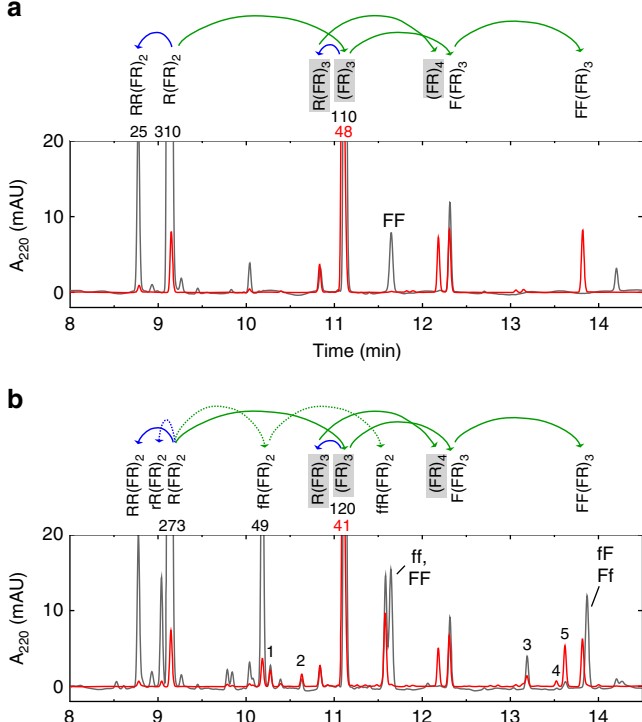

**Fig. 4** Sequence-specific and stereospecific, consecutive triple addition reactions with the R(FR)$_2$/(FE)$_4$ mixture. Reverse-phase HPLC chromatograms of the addition reactions of a mixture of arginine and phenylalanine (**a**), and a mixture of DL-arginine and DL-phenylalanine (**b**) with substrate R(FR)$_2$, in presence (red) or absence (black) of template (FE)$_4$. The maximum absorbance of the peaks that go off scale are listed above the peaks and the overlaid chromatograms have been scaled to represent the same amount of the total reaction. The substrate and template concentration was 100 μM, and amino acids were all at 200 μM of each enantiomer. The sequential additions of amino acids are indicated by the curved arrows: green for phenylalanine and blue for arginine, dotted lines are for D-enantiomers and lower case letters indicate the positions of D-residues. The isotactic and sequence-specific products are highlighted in gray. The labeled products were identified by mass spectrometry and control reactions with just D-enantiomers. The numbered products, identified only by mass spectrometry (sequence and stereochemistry not identified) have the following compositions: 1. R$_4$F$_3$, 2. R$_4$F$_3$, 3. R$_3$F$_4$, 4. R$_3$F$_5$ 5. R$_3$F$_4$

fact that the (FR)$_3$/(FE)$_4$ mixture forms an amyloid implies that the single phenylalanine addition product of R(FR)$_2$ would aggregate with (FE)$_4$. To probe the combined sequence and stereoselectivity of the soluble R(FR)$_2$/(FE)$_4$ mixture, we added both activated DL-phenylalanine and DL-arginine (each enantiomer at 200 μM). The precipitate from this reaction was collected at 100,000 g, solubilized in guanidine and analyzed by HPLC. Of the 64 possible triple addition products, only three could be detected in the insoluble fraction: two major products L-(FR)$_4$ and L-FF(FR)$_3$ and a minor product of undetermined stereo-chemistry, however most likely fFFRFRFR-NH$_2$ in which the final addition is a D-phenylalanine. (Fig. 4b). To help identify the numerous diastereomeric reaction products, the same reactions were carried out separately with D-amino acids or L-amino acids and also individually with arginine or phenylalanine. From these results, it was apparent that in the racemic reactions, the D-amino acids have little impact on the yield of the L-products. (Fig. 4a). The reaction with R(FR)$_2$ alone does not yield a precipitate and

the HPLC chromatogram of its reaction products has no detectable (FR)$_4$.

**Regioselectivity in the additions to (OV)$_4$ templated by V (DV)$_4$.** In order to address the relevance and general feasibility of this amyloid templating mechanism as a prebiotic reaction, we also designed an analogous substrate-template peptide pair from amino acids that are more likely to have been abundant in a prebiotic setting[25]. The valine/aspartate template V(DV)$_4$ (Ac-VDVDVDVDV-NH2) and valine/ornithine substrate (OV)$_4$ (OVOVOVOV-NH2) are, as with the FE/FR peptides, soluble in isolation and form an amyloid when mixed (Supplementary Fig. 9), yet they have an added complexity: the four ornithine residues and the N-terminus of (OV)$_4$ present a total of five addition-reactive sites. Indeed, in the standard conditions with two equivalents of activated valine, addition occurs equally well at all five sites of the soluble substrate, yielding the N-terminal addition product V(OV)$_4$ at 5%, constituting only 15% of the single addition products. With the (OV)$_4$/V(DV)$_4$ amyloid, the yield of V(OV)$_4$ is increased to 14%, constituting 65% of the single addition products (Fig. 5a). Taking into account the five potential reactive sites, the increase in N-terminal specificity in the amyloid aggregate compared to soluble peptide is about 10-fold. Since all five valine addition products are not resolved by HPLC, the elution time for V(OV)$_4$ was identified with an authentic sample and each HPLC peak analyzed by NMR (Supplementary Fig. 10). The yield and regioselectivity is even higher for phenylalanine for which N-terminal additions comprise 85% of single addition products (or about 23-fold more than each sidechain) with a yield of 20% (Fig. 5b).

**Robustness of the amyloid templating reaction.** The resilience of amyloids under harsh conditions is one of the characteristics that makes them interesting from a prebiotic perspective. The templating mechanism described above further supports the notion of the amyloid as a molecular scaffold that could have fulfilled the catalytic and genetic requirements of an early life-form. In fact, the (OV)$_4$/V(DV)$_4$ amyloid, with its putative pre-biotic composition, retains enhanced N-terminal addition activity from pH 5.6–8.6, and in salt concentrations from 0 to 4 M NaCl. Within these ranges, the N-terminal specificity increases with decreasing pH or increasing NaCl concentration, both at a small expense to V(OV)$_4$ yield (Supplementary Tables 3 and 4). The stabilizing effect of the amyloid was very pronounced in a reaction at 90 °C for 6 h in which the amyloid both retained N-terminal addition specificity and resisted hydrolysis while the soluble peptide yielded dozens of products as a result of hydrolysis, ornithine sidechain additions and multiple additions (Fig. 6).

While the reaction outcomes are shown to be influenced by many factors, both yield (of sequence-specific products) and specificity are consistently higher for the amyloid substrates compared to the soluble substrates. Also, in the course of the study we found that phosphate, the buffer used in all experiments except the NaCl series, is moderately inhibitory to the CDI activated additions, reducing the yields 2–3-fold at 50 mM compared to yields at 20 mM NaPO$_4$; however, it has no effect on the specificity. Of course, the equilibrium between the amyloid and soluble peptides will also affect the yield and specificity of the reactions. Under our standard aggregation conditions (100 μM substrate, 120–130 μM template, pH 7.4) we quantitated the amount of substrate that remains soluble for the three different amyloids studied. We could not detect soluble substrate peptide for the (FR/FE) amyloids, however about 3% of (OV)$_4$ remained soluble in a mixture with its template V(DV)$_4$.

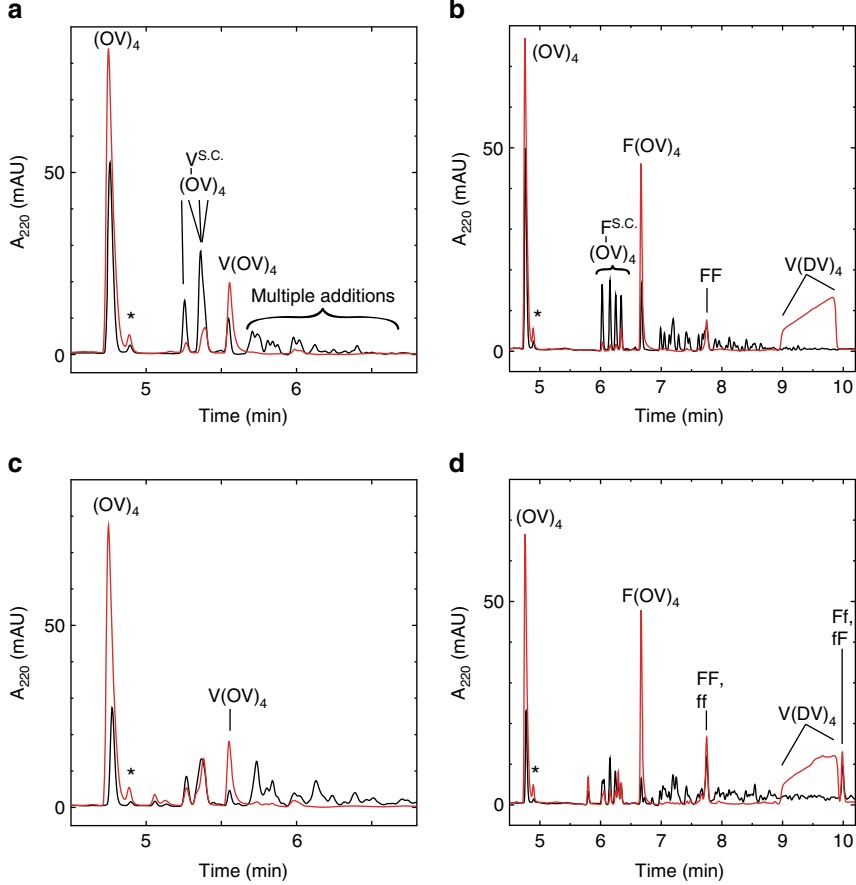

**Fig. 5** Regio-specific and stereo-specific addition reactions with the $(OV)_4/V(DV)_4$ amyloid. Reverse-phase HPLC chromatograms of the addition reactions with valine (**a**), phenylalanine (**b**), DL-valine (**c**) and DL- phenylalanine (**d**). Reactions with just substrate $(OV)_4$ are the black traces and those with the amyloid $(OV)_4/V(DV)_4$ are the red ones. The ornithine sidechain addition products are indicated with the superscript S.C. The * indicates the C-terminal deamidated substrate that remained after purification. The $(OV)_4$ concentration was 100 µM, the $V(DV)_4$ concentration was 120 µM and amino acids were all at 200 µM of each enantiomer. The addition product to the N-terminus was identified via NMR and reverse-phase HPLC analysis of an authentic V $(OV)_4$ control peptide (see Supplementary Fig. 10)

In both the $R(FR)_3/(FE)_4$ and $(OV)_4/V(DV)_4$ amyloid systems, multiple addition products are either absent or significantly reduced compared to their non-templated reactions. Likewise, there are double but not triple additions to the $(FR)_3/(FE)_4$ amyloid. Thus, it appears that the amyloids are inherently selective for the extension of substrates with a recessed N-terminus, that is, an N-terminus that does not extend to the end of the neighboring strands in the β-sheet. Although there is no high-resolution structural data on which to build a model, the FTIR spectra do indicate parallel β-sheets for the OV/DV system and anti-parallel β-sheets for the FR/FE system (Supplementary Figs. 1, 2 and 9). Therefore, we conclude that the inherent repetitive structures of the amyloid are generally well-suited to templating polymer addition reactions.

The concept of an amyloid as template for chemical reactions is not new. Nature has, in at least one known instance, taken advantage of the repetitive cross-β structure for the templated synthesis of the pigment melanin[26]. Just recently, an amyloid of the small peptide Ac-KLVFFAL-NH2 was designed that achieves template-directed polymerization of 6-amino-2-naphthaldehyde[7]. In terms, of self-replicative templating activities, amyloids have been shown to increase the rate of condensation of two halves of an amyloidogenic peptide via native chemical ligation[14,15]. The presented data, building upon these findings, greatly expands the prebiotic repertoire of amyloids by demonstrating sequence-selectivity, stereoselectivity and regioselectivity at the level of

amino acids. Since the sequential addition of amino acids to a peptide in a self-replicative manner is an indisputable prerequisite for life as it developed on Earth, these results are of particular interest to studies on life's molecular origins.

## Methods

**Peptide manipulations.** Synthetic peptide templates and substrates (GLS China) were synthesized using standard Fmoc chemistry on a Rink amide resin. The crude peptides were purified by reverse-phase HPLC on a Kinetex C18 5 µm 10 × 250 mm column (Phenomenex) in CH₃CN/H₂O/TFA or, for $V(DV)_4$ and $(FE)_4$ in CH₃CN/TEAA solvent systems. The purified peptides were quantitated by their calculated extinction coefficient at 214nm[27] and for phenylalanine-containing peptides also via total acid hydrolysis in 6 M HCl for 1 h at 160 °C followed by HPLC quantitation of phenylalanine. Pure peptides were stored in lyophilized aliquots. Before use, the peptides were dissolved in 50 mM Sodium phosphate buffer pH 7.4 except where specifically noted otherwise.

**Fibrillization.** Lyophilized aliquots of pure peptides were solubilized in 50 mM Sodium phosphate buffer pH 7.4 (20 mM for reactions in Fig. 1, Supplementary Table 1). Peptide co-aggregates were made simply by mixing substrate peptide (100 µM) and a small excess of template peptide (120–130 µM) and incubating them in a thermomixer (Eppendorf) with agitation at 800 r.p.m. and 37 °C. Fibrillization was monitored by centrifugation and circular dichroism spectroscopy and appeared to reach equilibrium within a few hours. The FR/FE formed flocculent aggregates and were briefly sonicated (10 s, 20% power, Bandelin Sonoplus HD 2070 with MS73 microtip) in order to improve liquid handling before performing the addition reactions. The amount of peptide substrates that remained soluble in the co-aggregate mixture was quantitated by centrifuging at 100,000 × g and analyzing the supernatant by HPLC. For the NaCl and pH screen, the co-

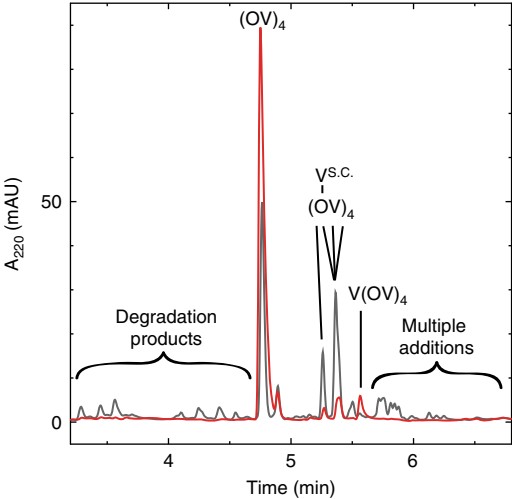

**Fig. 6** Hydrolytic resistance of the $(OV)_4/V(DV)_4$ amyloid. Reverse-phase HPLC chromatograms of the addition reactions with valine carried out at 90 °C for 6 h. The reactions with just the substrate $(OV)_4$ are the black traces and those with the amyloid $(OV)_4/V(DV)_4$ are the red ones. The $(OV)_4$ concentration was 100 μM, the $V(DV)_4$ concentration was 120 μM and valine concentration was 200 μM

aggregates were made as before at pH 7.4 in sodium phosphate buffer, collected by centrifugation and then resuspended in the new buffer/salt condition.

**Amino-acid addition reactions and analyses**. Amino acids were purchased with a purity of ≥98% and stock solutions were made in pure water except for aspartate, which was pH adjusted to ~ 7.0 with NaOH. Amino acids were activated with two equivalents of carbonyldiimidazole (CDI, ≥97% Aldrich) added on ice. For the phenylalanine-arginine mixtures, the amino acids were mixed before activation and for all reactions, the individual amino-acid concentrations were at least 20 mM except for tyrosine (2 mM) and tryptophan (10 mM) due to their low solubility. After 2 min on ice, the activated amino-acid solutions were diluted into the peptide mixtures at room temperature. After at least 18 h, the reactions were diluted with 3 volumes 8 M guanidine HCl (high purity, Pierce) in order to solubilize the peptides for reverse-phase analysis. A contaminant in the guanidine eluted close to some of the product peaks and so the 8 M solution was further purified by passing it over a C18 solid phase extraction column (Supelco). Reverse-phase analyses were performed on two different columns: 2.6 μm 4.6 × 150 mm Kinetex C18 column (Phenomenex) and 2.6μm 4.6 × 250 mm Aeris C18 column (Phenomenex) connected to an Agilent 1200 HPLC system equipped with an auto sampler and diode array detector. The soluble reaction products were injected onto the column and resolved using a linear gradient of acetonitrile with 0.1% TFA (10–40%, for FR/FE peptides and 10–30%, for VO/VD peptides) at a flow rate of 1–1.5 ml/min. While the experiments were not systematically carried out in replicates, many reactions were repeated several times and found to be reproducible. For example, the reaction with 100 μM valine in the $(OV)_4/V(DV)_4$ system was performed many times as a control. The average and standard deviation of seven fully independent (different days, different peptide aliquots) measurements of the yield and specificity (expressed as yield relative to sidechain products) for valine addition are as follows: $(OV)_4/V(DV)_4$ N-terminal specificity 67% (s.d. = 3.4%), yield 7.5% (s.d. = 0.79%), $(OV)_4$ N-terminal specificity 17% (s.d. = 3.3%), yield 2.8% (s.d. = 0.75%). Within one set of experiments that use the same fibril preparation, the variability will be even lower. Also, the reproducibility of the HPLC analyses themselves are high with typically lower than 1% variation in repeated injections.

**Kinetic data collection and analysis**. The reactions were performed as before with the following modifications: the phenylalanine was activated at 50 mM and then immediately before setting up the reaction, diluted with water to a concentration 10-fold higher than the desired reaction. This way, all reactions started with the same dilution factor of activated phenylalanine and the activation was always performed at the same concentration. After a short pulse on a vortexer to start the reaction, 20 μl aliquots were removed at different times and quenched by adding them to an HPLC auto sampler vial containing 60 μl 8 M guanidine, 45 mM HCl and vortexing. Twenty microliters of each diluted aliquot was injected onto the Kinetex C18 column as done previously. The peak area for the addition products were plotted vs. time and fit to a line, including a point for $t = 0$ s with no product. Some of the time points of the low concentration reactions with soluble peptide had no detectable product and these points were not included in the fitting. The slope of the fitted lines (initial rates, Supplementary Fig. 8a, c, e, g) were plotted vs.

phenylalanine concentration and fit to a line (Supplementary Fig. 8b, d, f, h), the slope of which would be the (pseudo) first order (with respect to phenylalanine) rate constant with the unit per second.

**Mass spectrometry**. HPLC-purified peptide samples were dried under vacuum and dissolved in 50% $CH_3CN$ with 0.05% formic acid. Mass spectra (ESI-QTOF) were collected on a Bruker maXis and calibrated with Tuning Mix (Agilent). The observed masses were all within 5 p.p.m. of the calculated masses (Supplementary Fig. 11).

**Circular dichroism and Fourier transform infrared spectroscopy**. The peptides and peptide mixtures (100 μM each peptide) were analyzed by circular dichroism spectroscopy on a Jasco J-815 with 1 mm path length and scanned from 260-190 nm at 50 nm/min, 1 nm band-pass, 8 s integration, averaged over 3 repetitions (except as noted in Supplementary Fig. 1e). The same samples were centrifuged at 25,000 × g, the insoluble material resuspended in 2 μl buffer and applied to a diamond ATR cell on a Bruker Alpha FTIR spectrometer. The samples were air-dried before measuring their spectra with 32 scans and a resolution of 2 cm$^{-1}$. For the soluble peptides and peptide mixtures, 10 μl of the solutions (100 μM substrate/130 μM template) were applied to the cell in 2 μl aliquots and allowed to dry between additions.

**Transmission electron microscopy**. The samples prepared as for CD spectroscopy were applied directly to glow-discharged carbon-coated copper grids, washed and then stained with uranyl formate. Images were recorded on a FEI Morgagni 268 electron microscope.

**NMR spectroscopy**. Since the substrate peptide $(OV)_4$ has five reactive sites for amino-acid addition (four ornithine side chains and the N-terminus) it can form five isomeric products. Simple mass spectrometry cannot differentiate the isomers so we employed 2D HMQC experiments to identify the valine addition reaction products in the HPLC chromatograms. In order to assign the HN and N chemical shifts of the valine in the N-terminal addition product, we synthesized and purified an authentic $V(OV)_4$ and measured its natural abundance spectrum in deuterated DMSO with 1% TFA at a concentration of 5 mM. An addition reaction with activated $^{15}$N-valine and the $(OV)_4$ substrate was performed and the products purified by HPLC. The NMR spectra of each of the three main HPLC peaks was measured at 20 μM peptide concentration so that only the isotope enriched valine was detected (Supplementary Fig. 10).

**Data availability**. The data that support the findings of this study are available from the authors on reasonable request.

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

## Acknowledgements

We thank the ETHZ ScopeM microscopy center and Pratibha Kumari for help with the EM images and Witek Kwiatkowski for many helpful insights and a critical reading of the manuscript. This work was supported by an ETH grant.

## Author contributions

S.K.R. performed the experiments. All authors contributed to the design of the experiments and J.G. and S.K.R. did the data analysis and wrote the manuscript.

## Additional information

**Competing interests:** The authors declare no competing financial interests.

