## [Peer Review File · Nature Communications]

Reviewers' comments:

Reviewer #1 (Remarks to the Author):

This is a very interesting manuscript on the templating of amino acid polymerization by an amyloid scaffold. The authors provide a very experimental demonstration to the theory for the roles of amyloids in pre-biotic peptide replication and subsequently the origin of life. This is a very appealing hypothesis as the current RNA-based origin of life cannot explain the early stages before complex molecules as polynucleotides could emerge. Moreover, as correctly stated by the authors, the stability of amyloids is much higher as compared to RNA. The manuscript is well written and conclusions are being supported by the experimental results. Here are few points to consider:

1. The authors report that the yield of the phenylalanine addition product was much higher in the aggregated sample as compared to the soluble control. Could it be related to the ability of phenylalanine to self-associate as was reported in the past?

2. The observed stereoselectivity is quite intriguing. What would be the molecular mechanism for achieving such notable selectivity? Does the template bind only one of the enantiomer or actually the catalytic product with the different enantiomers is less stable? Furthermore, what is known about the stability and the ability of diastereomeric peptides to self-assemble?

3. The authors state: "we also designed an analogous substrate-template peptide pair from amino acids that are more likely to have been abundant in a prebiotic setting" – what is actually the predicted amino acid distribution in the prebiotic stage?

Minor point:

1. References #6 was already published: Nature Chemistry 9, 805–809 (2017)

Reviewer #2 (Remarks to the Author):

Review on Greenwald and coworkers "Sequence- and Stereo-Selective Amyloid-Templated Amino Acid Polymerization"

The paper describes the polymerization of short peptides templated by cross- β -fibril 'sandwich' assemblies. The idea is nice as it is based on the formation of non-ideal sheets by 8/7 aa chains and templating the addition of an incoming aa due to structure and charge matching with existing vacancies. The templating effect is very nicely demonstrated(!) as it brings about sequence specificity, chiro-selectivity (L vs. D), and regioselectivity (in binding to ornithine). This discovery justifies publication in Nature Communications.

I would like however to point out several major issues and more minor aspects that prevent me from recommending publication of the paper in its current form.

1. The introduction section (or intro to specific assays) missed citation of a large body of work very relevant to the current paper.

i. The question "...whether amyloids can replicate themselves using simple chemical building blocks." (p. 2) has been addressed several times before with related systems (e.g., by Lynn, Ashkenasy, Otto...). The very short reference to some of these works at the end of the paper does not reflect their contribution correctly.

ii. The self-assembly of EFn peptides (and related sequences) into fibril architectures has been studied quite extensively, first by Zhang and Rapaport, and later on by others. Some

characteristics of these structures should/can be given already at the intro part.

iii. Most important: the remarkable work on isotactic polymerization using racemic β -Sheets as templates, studied about a decade ago by Luisi and Lahav, was ignored altogether!

2. The structural characterization of the fibril assemblies formed by the peptides is very limited and as such not too-informative with respect to the main questions of this study. I also have reservations regarding some of the structural data observations.

i. The assembly formed by mixtures of FE4-R(FR)₃ are characterized by FTIR and CD (Fig. s1 a-d). It is much surprising to see that FE4 on its own does not form fibrils. The FEn peptides in various lengths have been previously shown to form fibrils under very similar concentrations and pH conditions (the referee is less familiar with the assembly of FRn type peptides). Have the authors used any specific method for solubilizing the compounds? Any unconventional equilibration procedure? Are they sure that the supernatant after sonication-centrifugation contained the designated peptide amount? It was also not clear to me what exact concentrations were used in the measurement; since fibril assembly sometimes depends critically on concentrations, for proper comparison, the spectra obtained for 100 micM FE4 + 100 micM R(FR)₃ should be compared to the spectra of 200 micM of FE4 alone (or R(FR)₃ alone).

ii. The FTIR data in Fig. s1a shows a very tiny peak at 1694 cm⁻¹. This peak is usually more pronounced for anti-parallel b-sheets; can this mean a different structure interpretation to the data?

iii. The authors can probably agree that the TEM images is quite poor in resolution. It seems like the mixtures form structures that are larger than single fibrils. Is that correct? Does this reflect bundling, or wider fibrils as observed for the FF type aggregates? In any case, cryo-EM measurements would yield images of structures much more relevant to the assemblies formed in solution; likewise, microscopy images of the FE4 and FR4 peptide assemblies alone should also be given as control (and complementary to the CD and FTIR measurements discussed in i).

3. It is hard for me to recommend the acceptance of a paper on catalysis/templating effects to a high-profile journal without any kinetic data. It is suggested that, as a minimum, the authors repeat the basic polymerization assays shown in figure 1 and collect data over time. The analysis of this data can probably provide insight into various mechanistic issues relevant to their study: Is the reaction 1st order, meaning that a preformed fibril is 'waiting' for an incoming aa, or it is 2nd order with respect to the growing chain and aa? Does the reaction go through different phases where the incoming aa react faster and slower? What is the rate limiting step of these reactions? Is steric (and charge matching) indeed a kinetic factor affecting the chain growth or it is more related to the formation of less or more well-ordered architectures? Etc.

4. I think that more in-depth explanation is needed for why higher stereo-selectivity is observed for incorporation of V versus that of F (Fig. s3).

5. For the data on stereoselectivity (Fig. 2 etc.), one would expect to see at least one time the control reaction of f (D-Phe) alone as the incoming ligand.

6. Peak assignment on fig. 4 is quite difficult and thus nice. The authors can/should provide the raw MS and HPLC data in the SI.

7. Minor: the term amyloid is frequently used in biology (e.g., for the A β structures) but in peptide chemistry most people would prefer to use 'fibrils'. Please consider if this can be changed in the title and along the paper.

8. Minor: as a reader, I would like to have some technical experimental data next to a discussed figure, as to help understanding what exactly is presented. The authors can consider move or duplicate some of their methods into the figure captions.

Reviewer #3 (Remarks to the Author):

The idea that peptides, acting as both an informational and catalytic entity, preceded life on Earth is fascinating and appeals to this referee much more than the RNA-world hypothesis.

Selfreplicating amyloids are an attractive possible requirement for this peptide-world hypothesis. The authors investigate well-designed oppositely charged peptide templates and substrates [(FE)₄ and R(FR)₃] and find that within the resulting amyloid, the substrate is extended by the activated phenylalanine to give (FR)₄. Remarkably, careful control experiments show that the amyloid is sequence specific for the addition of hydrophobic amino acids, enhancing only single additions to the substrate. These results are fascinating and should be published in Nature Communications as they are highly relevant to chemists and biologists alike.

However, revisions are needed. The authors suddenly state that "Thus, the reaction is steered by the stereochemistry of the amyloid rather than simply...". How do they come to this conclusion? The logical connection of this statement to the results eludes this referee. Also, a serious issue that needs to be addressed by the authors is the stereochemical description. The reaction products are not enantiomers but, as they correctly state, diastereomers. Therefore, e.e. values are not appropriate but d.e values. This must be carefully corrected in the manuscript and SI. In this context, it would be very interesting to actually measure the e.e.'s of the remaining amino acids. An enrichment in the D-amino acid is expected.

In summary, this is a very interesting manuscript, supporting a fascinating hypothesis on the origin of life. The authors need to carefully reevaluate the stereochemistry of the studied processes and the use of stereochemical terms.

Minor corrections:

"Table 1 Select peptides" should be "Table 1 Selected peptides"

Title and throughout manuscript: "Stereo-Selective" should be "Stereoselective"

In SI "Stereospecific" is used. It means something else and should not be used here. The correct term is "stereoselective".

Response to Reviewers' comments:

Reviewer #1

This is a very interesting manuscript on the templating of amino acid polymerization by an amyloid scaffold. The authors provide a very experimental demonstration to the theory for the roles of amyloids in pre-biotic peptide replication and subsequently the origin of life. This is a very appealing hypothesis as the current RNA-based origin of life cannot explain the early stages before complex molecules as polynucleotides could emerge. Moreover, as correctly stated by the authors, the stability of amyloids is much higher as compared to RNA. The manuscript is well written and conclusions are being supported by the experimental results. Here are few points to consider:

1. The authors report that the yield of the phenylalanine addition product was much higher in the aggregated sample as compared to the soluble control. Could it be related to the ability of phenylalanine to self-associate as was reported in the past?

The idea raised by the reviewer would be an interesting possibility; however, the addition product is also increased for valine as well as arginine (in the reaction with **(FR)₃**) and thus we did not pursue it.

2. The observed stereoselectivity is quite intriguing. What would be the molecular mechanism for achieving such notable selectivity? Does the template bind only one of the enantiomer or actually the catalytic product with the different enantiomers is less stable? Furthermore, what is known about the stability and the ability of diastereomeric peptides to self-assemble?

It is expected that the chirality of the amyloid as well as the steric requirement that peptides in beta-sheets be isotactic, influence in the observed stereoselectivity. In the original manuscript, we had hypothesized that there is an interaction between the incoming activated amino acid and the amyloid template. In the revised manuscript, we show now that the reaction kinetics for the addition of L versus D phenylalanine indicate distinct mechanisms, however we are not yet able to provide a detailed reaction mechanism.

The stability of the diastereomeric products are both stable enough that their stabilities should not influence the product yield. As referenced in the original manuscript, it has been shown that non-isotactic peptides do not assemble into beta structures unless there is a minimum stretch of isotactic sequence.

3. The authors state: "we also designed an analogous substrate-template peptide pair from amino acids that are more likely to have been abundant in a prebiotic setting" – what is actually the predicted amino acid distribution in the prebiotic stage?

While there is no definitive answer to what is a prebiotic amino acid, there is some consensus on those that are more likely to have been available in a prebiotic environment. We have included a reference to Pudritz *et al.* for the studies that address this issue.

Minor point:

1. References #6 was already published: *Nature Chemistry* 9, 805–809 (2017)

Thank you for this input. The reference has been corrected.

Reviewer #2

Review on Greenwald and coworkers "Sequence- and Stereo-Selective Amyloid-Templated Amino Acid Polymerization"

*The paper describes the polymerization of short peptides templated by cross- β -fibril 'sandwich' assemblies. The idea is nice as it is based on the formation of non-ideal sheets by 8/7 aa chains and templating the addition of an incoming aa due to structure and charge matching with existing vacancies. The templating effect is very nicely demonstrated(!) as it brings about sequence specificity, chiro-selectivity (L vs. D), and regioselectivity (in binding to ornithine). This discovery justifies publication in *Nature Communications*.*

I would like however to point out several major issues and more minor aspects that prevent me from recommending publication of the paper in its current form.

1. The introduction section (or intro to specific assays) missed citation of a large body of work very relevant to the current paper.

i. The question "...whether amyloids can replicate themselves using simple chemical building blocks." (p. 2) has been addressed several times before with related systems (e.g., by Lynn, Ashkenasy, Otto...). The very short reference to some of these works at the end of the paper does not reflect their contribution correctly.

We agree with the concerns raised by the reviewer. The original manuscript was written for a shorter format as a letter in *Nature* and then forwarded directly to *Nat. Comm.* The introduction has been expanded now to include more of the previous studies on amyloid replicating systems.

ii. The self-assembly of EFn peptides (and related sequences) into fibril architectures has been studied quite extensively, first by Zhang and Rapaport, and later on by others. Some characteristics of these structures should/can be given already at the intro part.

The most relevant characteristic (binary alternating sequence of hydrophobic and hydrophilic residues) was referenced in the original manuscript. We would like to thank the reviewer to mention the work by Rapaport, which has now been added as a reference in the introduction and legend of Fig. S1.

iii. Most important: the remarkable work on isotactic polymerization using racemic β -Sheets as templates, studied about a decade ago by Luisi and Lahav, was ignored altogether!

We apologize for this oversight, which has been corrected in the revised manuscript. Thank you for pointing it out.

2. The structural characterization of the fibril assemblies formed by the peptides is very limited and as such not too-informative with respect to the main questions of this study. I also have reservations regarding some of the structural data observations.

i. The assembly formed by mixtures of FE4-R(FR)3 are characterized by FTIR and CD (Fig. s1 a-d). It is much surprising to see that FE4 on its own does not form fibrils. The FEn peptides in various lengths have been previously shown to form fibrils under very similar concentrations and pH conditions (the referee is less familiar with the assembly of FRn type peptides). Have the authors used any specific method for solubilizing the compounds? Any unconventional equilibration procedure? Are they sure that the supernatant after sonication-centrifugation contained the designated peptide amount? It was also not clear to me what exact concentrations were used in the measurement; since fibril assembly sometimes depends critically on concentrations, for proper comparison, the spectra obtained for 100 micM FE4 + 100 micM R(FR)3 should be compared to the spectra of 200 micM of FE4 alone (or R(FR)3 alone).

We were aware of the aggregation-prone nature of both **(FE)₄** and **R(FR)₃** and so we were careful to ascertain that the isolated peptides were in fact soluble under the reaction conditions. We have included more data that speaks to this conclusion (Figs. S1 and S2). Now we have the CD spectrum recorded at 1 mM (10x the concentration used in the study) as well as at two lower pH values (4 and 6). At pH 6 the CD spectrum is very similar to the pH 7.4 spectrum and only at pH 4 does the sample become visibly precipitated and its CD spectrum more typical of a beta structure. Also, we prepared an EM grid with a “soluble” pH 7.4 **(FE)₄** sample at 450 μ M and found only small amount of amorphous material in the EM images. However, EM images of **(FE)₄** at pH 4 reveal amyloid-like fibrils. These results are consistent with previous findings with a similar peptide: PE(FE)₄P (Rapaport et al. *JACS* **122**, 12523-29 (2000), which is referenced now in the manuscript). This peptide becomes surface active only below pH 7.2 and in fact in its aggregated form has a weak IR band at 1694 cm^{-1} which was attributed to an anti-parallel arrangement.

There was no special handling of the peptides apart from what was described in the manuscript. The reason for the sonication step was to increase the reproducibility of pipetting the slightly flocculent aggregate. We had not in fact checked the effect of sonication on solubility before submitting the original manuscript but at the suggestion of the reviewer we did do this. We found no “solubilization” of the peptides caused by the brief sonication, however, we did see a small increase in yield with sonication that is probably due to simply increasing the accessible surface area of the aggregates. These sonication results are not included in the manuscript as we do not find them relevant, however we can add them to the supplemental material if requested to do so.

ii. The FTIR data in Fig. S1a shows a very tiny peak at 1694 cm^{-1} . This peak is usually more pronounced for anti-parallel β -sheets; can this mean a different structure interpretation to the data?

The reviewer is correct that the peak at 1694 cm^{-1} that we observed for the **R(FR)₃/FE₄** and **(FR)₃/FE₄** samples is small, however, in our experience its size is typical for what many researchers attribute to anti-parallel β structure (including the Rapaport reference above). Another example shown below is from a recent review on using FTIR as a tool to study amyloids (*Biochimica et Biophysica Acta* 1828 (2013) 2328–2338)

Even a theoretical (calculated) spectra for an infinite anti-parallel β -sheet has a 10-fold smaller intensity at 1695 cm^{-1} compared to that at 1630 cm^{-1} (*Biopolymers* 15 (1976) 607-625). In any case, the precise nature of the β -sheet is not highly relevant to our results, as can be inferred from Figure 1a, for which no directionality of the β -strands is given.

iii. The authors can probably agree that the TEM images is quite poor in resolution. It seems like the mixtures form structures that are larger than single fibrils. Is that correct? Does this reflect bundling, or wider fibrils as observed for the FF type aggregates? In any case, cryo-EM measurements would yield images of structures much more relevant to the assemblies formed in solution; likewise, microscopy images of the FE4 and FR4 peptide assemblies alone should also be given as control (and complementary to the CD and FTIR measurements discussed in i).

The EM images (Figs. S2 and S9) have been kept at the original resolution (1376 x 1032 pixels) and on our computer screens they are about as clear as EM images come. Perhaps some resolution was lost in the transfer of the file from the *Nature Communications* editors? The **R(FR)₃/ (FE)₄** fibers are rather typical amyloid fibers. The bundling is likely to occur in solution (they are large aggregates much of it settles out of solution over the period of a few hours), however as the reviewer pointed out, the negative staining procedure could lead to artifacts such as bundled fibers. The **(FR)₃/ (FE)₄** fibers are significantly different with one dimension being much longer (wider fibers) with a visible twist to some of the fibers.

We do not see the need for cryo-EM data for this manuscript as more detailed structural information is not critical for the scope of the work presented. We did however measure and include the EM image for the soluble and a low pH aggregated form of **(FE)₄** in Fig. S2.

3. It is hard for me to recommend the acceptance of a paper on catalysis/templating effects to a high-profile journal without any kinetic data. It is suggested that, as a minimum, the authors repeat the basic polymerization assays shown in figure 1 and collect data over time. The analysis of this data can probably provide insight into various mechanistic issues relevant to their study: Is the reaction 1st order, meaning that a preformed fibril is 'waiting' for an incoming aa, or it is 2nd order with respect to the growing chain and aa? Does the reaction go through different phases where the incoming aa react faster and slower? What is the rate limiting step of these reactions? Is steric (and charge matching) indeed a kinetic factor affecting the chain growth or it is more related to the formation of less or more well-ordered architectures? Etc.

We wholly agree that some basic kinetics could shed light on the mechanism and so we have added this to the manuscript: In addition to the suggested time course (24 h at the standard reaction conditions: 100 μ M peptides and phenylalanine), we also measured the initial rate of the reaction for the amyloid and the soluble peptide with L and D phenylalanine at a range of phenylalanine concentrations (25-5000 μ M). The results highlight the stereoselectivity and the differences between the templated and non-templated reactions. While the non-templated reactions are not surprisingly of first order, we see that the templated L-phenylalanine additions do not follow a simple 1st order mechanism. For the analysis of the kinetics we have added the following text in the manuscript and Supplemental information accompanied with several Figures (Figure S5-S8):

Main Text

“For an initial assessment of the reaction kinetics, we measured the time dependence of product formation at our standard conditions (100 μ M activated phenylalanine) over a period of 24 h. The results depicted in Fig. S5 show that the majority of the **(FR)₄** product is formed within 30 min, with the yield peaking around 3 hours and then sinking about 20%, primarily due to multiple additions. In an effort to get a more detailed understanding of the reaction mechanism we measured the early kinetics of the reaction for both **R(FR)₃/(FE)₄** and the soluble **R(FR)₃** at various L- and D-phenylalanine concentrations (Figs. S6-S8). The additions of L- or D-phenylalanine to soluble peptide appear to be 1st order with respect to phenylalanine (Fig. S8c-d) as would be expected for a simple uncatalyzed addition. In contrast, the reaction kinetics of the L-additions to the **(FR)₄/(FE)₄** amyloid is much faster and of greater complexity, including a burst phase (Figs. S7 and S8a). The data fit neither a 1st order reaction nor a Michaelis Menten-type mechanism (assuming a turn-over of 1). This complexity is specific for L-additions, since the additions of D-phenylalanine to **(FR)₄/(FE)₄** appear to be 1st order with respect to phenylalanine (Fig. S8b) (for some further details we refer the reader to the Supplementary Information).”

Supplemental Information

“On the complexity of the kinetics of L-Phenylalanine addition to the R(FR)₃/(FE)₄ amyloid

To better characterize the reaction, we measured the early kinetics of the reaction for both $R(\mathbf{FR})_3/(\mathbf{FE})_4$ and the soluble $R(\mathbf{FR})_3$ at various L- and D-phenylalanine concentrations (Figs. S6-S8). In contrast to the additions of L- or D-phenylalanine to soluble peptide the initial rate of L-additions to $(\mathbf{FR})_4/(\mathbf{FE})_4$ did not appear to be 1st order with respect to phenylalanine (Fig. S8a). This finding indicates a complex mechanism, which is not unexpected considering that the observed concentration dependence of the stereoselectivity implies some binding interaction between the substrates before the reaction occurs (Fig. 2). It is reasonable to expect that the stability of the amyloid would inhibit the exchange of peptides on the time scale of the reaction and therefore preclude a truly catalytic mechanism. However, assuming a turn-over of 1, the initial rate of the amyloid reaction could be expected to follow Michaelis-Menten kinetics. We attempted to fit the initial rates to a Michaelis-Menten model but this also yielded only poor fits. Furthermore, the $(\mathbf{FR})_4/(\mathbf{FE})_4$ reaction with L- but not D-phenylalanine appears to have a burst phase in the first few minutes of reaction, clearly visible in the 25, 100 and 200 μM reactions but absent in the D-additions of similar initial rate (at 1000, 2000 and 5000 μM D-phenylalanine Fig. S7). This is unlikely to be analogous to an enzyme burst rate because multiple turnover is not expected. Thus, no model for the L-phenylalanine additions to $(\mathbf{FR})_4/(\mathbf{FE})_4$ can be put forward yet. This uncertainty is due to a number of factors: (i) The initial rate at 500 μM phenylalanine and above was too fast to be accurately measured. (ii) The activation of the amino acid involves at least one long-lived intermediate before the formation of the more reactive species (most likely N-carboxyanhydride)¹, and the concentrations of these species and the kinetics of their interconversion. (iii) The amyloid may structurally rearrange upon reaction, both locally by stabilizing the β -sheet (potentially influencing the neighboring reactive sites) as well as on the mesoscopic scale by enhancing protofilament-protofilament interactions (altering the accessibility of active sites). (iv) Polymorphisms may be present as often observed for amyloids² and these polymorphisms may exhibit distinct activities. In summary, while the kinetic data remain inconclusive concerning a model for the reaction mechanism, it does point to different mechanisms for the L-phenylalanine versus D-phenylalanine additions to the amyloid additions and all additions to the soluble peptide.”

In summary, although we followed the suggestion of the reviewer with some basic kinetic measurements complemented by measurements over a large concentration range, detailed answers to the questions posed by the reviewer were not revealed. While such mechanistic details would be nice to have, they are not critical for the presented work.

4. I think that more in-depth explanation is needed for why higher stereo-selectivity is observed for incorporation of V versus that of F (Fig. s3).

The stereoselectivity of the amyloid is in fact rather similar for valine and phenylalanine considering the large difference when compared to the soluble peptide. However, we see how the numbers presented in Table S2 could be taken as a quantitative measure of the stereoselectivity when they were meant to be a qualitative measure. A note to this effect has been added in for the reference to Table S2 (in the legend to Fig. 2). More striking than the valine versus phenylalanine enantioselectivity is the concentration dependence of this selectivity, which appears to be larger for phenylalanine.

5. For the data on stereoselectivity (Fig. 2 etc.), one would expect to see at least one time the control reaction of f (D-Phe) alone as the incoming ligand.

The reaction with D-phenylalanine alone is now contained in the kinetic analysis (Figs. S6-S8).

6. Peak assignment on fig. 4 is quite difficult and thus nice. The authors can/should provide the raw MS and HPLC data in the SI.

This data is now presented in Fig. S11

7. Minor: the term amyloid is frequently used in biology (e.g., for the A β structures) but in peptide chemistry most people would prefer to use 'fibrils'. Please consider if this can be changed in the title and along the paper.

We are sensitive to the various meanings of amyloid but prefer to use this term over fibril because it has a more specific meaning: cross-beta fibrils. Since amyloid is a shorter version that contains this specific information, it is still useful.

8. Minor: as a reader, I would like to have some technical experimental data next to a discussed figure, as to help understanding what exactly is presented. The authors can consider move or duplicate some of their methods into the figure captions.

We followed the suggestion of the reviewer in part. We are happy to add further technical data in the figure captions on the request of the editor.

Reviewer #3

The idea that peptides, acting as both an informational and catalytic entity, preceded life on Earth is fascinating and appeals to this referee much more than the RNA-world hypothesis. Selfreplicating amyloids are an attractive possible requirement for this peptide-world hypothesis. The authors investigate well-designed oppositely charged peptide templates and substrates [(FE)₄ and R(FR)₃] and find that within the resulting amyloid, the substrate is extended by the activated phenylalanine to give (FR)₄. Remarkably, careful control experiments show that the amyloid is sequence specific for the addition of hydrophobic amino acids, enhancing only single additions to the substrate. These results are fascinating and should be published in Nature Communications as they are highly relevant to chemists and biologists alike.

However, revisions are needed. The authors suddenly state that "Thus, the reaction is steered by the stereochemistry of the amyloid rather than simply...". How do they come to this conclusion? The logical connection of this statement to the results eludes this referee.

We agree that the sentence was lacking in logic. We have removed it and added the following:

“Thus, the reaction on the amyloid is not simply enhanced by an increase in the nucleophilicity of the N-terminal amine, but is influenced by the physicochemical properties of the incoming amino acid.”

Also, a serious issue that needs to be addressed by the authors is the stereochemical description. The reaction products are not enantiomers but, as they correctly state, diastereomers. Therefore, e.e. values are not appropriate but d.e values. This must be carefully corrected in the manuscript and SI.

Thank you for this correction. It has been made

In this context, it would be very interesting to actually measure the e.e.'s of the remaining amino acids. An enrichment in the D-amino acid is expected.

We have done this measurement for a mixture of 100 μM (^{15}N)-L- and 100 μM D-Phenylalanine. The reactions were purified by HPLC and the phenylalanine peaks analyzed by ESI-MS. The results conform to the reviewer's (and our) expectations and are presented in Fig. S4

In summary, this is a very interesting manuscript, supporting a fascinating hypothesis on the origin of life. The authors need to carefully reevaluate the stereochemistry of the studied processes and the use of stereochemical terms.

Minor corrections:

"Table 1 Select peptides" should be "Table 1 Selected peptides"

It has been corrected.

Title and throughout manuscript: "Stereo-Selective" should be "Stereoselective"

We have made this correction except in phrases where multiple adjectives are strung together as in “regio- and stereo-selective”.

In SI "Stereospecific" is used. It means something else and should not be used here. The correct term is "stereoselective".

Thank you for catching this error. It has been corrected.

REVIEWERS' COMMENTS:

Reviewer #1 (Remarks to the Author):

The authors fully addressed my comments. I have no further comments or inquiries. I am happy to recommend the publication of this fine work.

Reviewer #2 (Remarks to the Author):

Greenwald and coworkers "Sequence- and Stereo-Selective Amyloid-Templated Amino Acid Polymerization"

The authors have gone a long way to upgrade their paper. Like the other two referees, this referee also thinks that the paper is interesting, well written, and suitable for publication in a high-profile journal.

Minor issues at the level of paper editing may also be done for better presentation:

1. In their answer to my comment 2/iii, the authors write "We do not see the need for cryo-EM data for this manuscript as more detailed structural information is not critical for the scope of the work presented.". While the paper is indeed inclusive in its current form, I would add to the main manuscript a note saying just the opposite, something like: 'further high-resolution structure analysis would be useful for better explaining the observed reaction mechanism'.

2. The new results showing the system's kinetic and stereo selectivity are very useful. Why not showing some of that in the main manuscript? Another comment here: a legend is missing for the different concentrations used in Fig. s6.

Reviewer #3 (Remarks to the Author):

All requested changes have been made. This manuscript is now acceptable.

Response to Reviewers' comments:

Reviewer #1 (Remarks to the Author):

The authors fully addressed my comments. I have no further comments or inquiries. I am happy to recommend the publication of this fine work.

We thank you for your helpful criticisms to the original manuscript.

Reviewer #2 (Remarks to the Author):

*Greenwald and coworkers "Sequence- and Stereo-Selective Amyloid-Templated Amino Acid Polymerization"
The authors have gone a long way to upgrade their paper. Like the other two referees, this referee also thinks that the paper is interesting, well written, and suitable for publication in a high-profile journal.*

We thank you for your helpful criticisms to the original manuscript

Minor issues at the level of paper editing may also be done for better presentation:

1. In their answer to my comment 2/iii, the authors write "We do not see the need for cryo-EM data for this manuscript as more detailed structural information is not critical for the scope of the work presented.". While the paper is indeed inclusive in its current form, I would add to the main manuscript a note saying just the opposite, something like: 'further high-resolution structure analysis would be useful for better explaining the observed reaction mechanism'.

We agree that detailed structural information would be helpful and may be necessary to get a detailed understanding of the reaction mechanism with the amyloid templates. We have added a concluding sentence to this effect in the Supplementary Discussion where the complexity of the kinetics is discussed.

2. The new results showing the system's kinetic and stereo selectivity are very useful. Why not showing some of that in the main manuscript? Another comment here: a legend is missing for the different concentrations used in Fig. s6.

We understand the reviewer's point but still prefer to leave the kinetic data in the Supplementary Information, near to the Supplementary Discussion. We feel that because the main conclusions are based on the templating effect, it could disrupt the flow of the main arguments if more kinetic data is added in the main text. However, we leave a final decision on the placement of the figures to the editor.

We have added the missing legend to Supplementary Fig. 6.

Reviewer #3 (Remarks to the Author):

All requested changes have been made. This manuscript is now acceptable.

We thank you for your helpful criticisms to the original manuscript.